# PREDICTING TARGETS WITH DATA FROM NON-CONFORMING SOURCES

**Alexander Capstick**
Imperial College London
`alexander.capstick19@imperial.ac.uk`

**Payam Barnaghi**
Imperial College London
`pbarnaghi@imperial.ac.uk`

## ABSTRACT

Machine learning applications to real-world settings are often tasked with making predictions on data generated by multiple sources. There are many methods for understanding when data is Out-Of-Distribution (OOD). A less explored area of importance is where OOD data can be considered In-Distribution (ID) when conditioned by its generating data source. Within this preliminary research, we focus on this issue and propose methods for building classification models capable of making predictions on data in which labels can depend on their source.

## 1 INTRODUCTION

In applications of machine learning to real-world settings, distinct data sources appear naturally. Personalised models (Tan et al., 2022) appear in Federated Learning, and there is extensive literature on the detection of Out-Of-Distribution data (Yang et al., 2021) and learning from noisy labels (Song et al., 2022), focused on rejection of data points that lead to reductions in performance on In-Distribution data. Here we focus on the setting in which data sources generate similarly distributed data points, with differing label assignments that we want to learn. Wiens et al. (2019) and Oh et al. (2018) discussed this context, in which variables associated with some risk in one hospital were protective in another. Additionally, Multi-Task Learning similarities are discussed in Appendix A.1.

Given a dataset, $D$, we consider that $D$ is composed of several sources, $S$ where $D = \bigcup_{i \in |S|} S_i$. Each data source, $S_i$, contains its own label assignment, such that a target for a data point depends on its source. Within this preliminary work, we consider all input data to be drawn from the same distribution, with only the label assignments depending on the source.

## 2 PROPOSED SOLUTIONS

To learn multiple label assignments, dependent on a point's underlying data source, we evaluated multiple methods [1]. **Input Append:** We include the source information in the input. **Classifier Append:** We append the source information to the output of a predictive model and apply another fully connected layer before the output. **Predict Mapping:** We use the source information to predict a matrix that maps the outputted layer from the predictive model to a source dependent output (Figure 1a). **Separate Classifier:** A separate classification layer is built per source (Figure 1b). **Calculate Mapping:** A binary mapping matrix is calculated per source, to allow for the mapping of the output from a predictive model to the source assigned labels, motivated by a change in basis of the label-spaces (Figure 1b). See A.4 for further information on these methods.

Predicting or calculating a matrix to perform a change of basis on the label-space is motivated by the following: consider a predictive model assigning a class, $c \in C$ to a data point. A source dependent label-space is equivalent to the space spanned by $C$ after one-hot encoding, under a change of basis. If the source dependent spaces contain the same number of dimensions, calculating the mapping between them can be done by calculating a change of basis matrix, i.e: for all labels in $C_{S_i} \in \mathbb{R}^{|C| \times |C|}$, labels in $C_{S_j}$ can be calculated using $C_{S_i} M_{S_j}^{S_i} = C_{S_j}$. Since $C_{S_i}$ can be written as the identity matrix by using one-hot encoding and rearranging columns, $M_{S_j}^{S_i} = C_{S_j}$. Therefore, to

---

[1]`https://github.com/alexcapstick/Non-Conforming-Sources.`

calculate the labels in $S_j$ based on $S_i$, $C_{S_j}$ should be ordered such that $C_{S_i}$ forms the identity matrix. However, since we do not know the label of a single data point under two different sources, we are unable to calculate $M_{S_j}^{S_i}$ directly. Instead, we attempt to calculate the change of basis between the predicted label from the final layer of the predictive model, and the source dependent labels.

This insight is used in "Predict Mapping" and "Calculate Mapping"; the former uses source information and gradient descent to learn a binary mapping, and the latter calculates mappings from predicted labels to expectations of the targets on each batch and source. "Separate Classifier" (linear layer for each source) mapping is similar, except it contains a bias and continuous valued matrix.

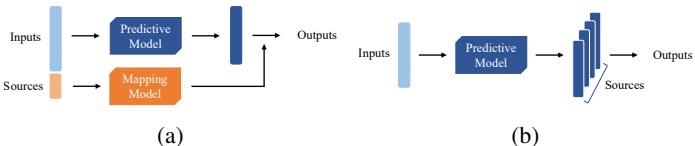

Figure 1: (a)"Predict Mapping" , (b) "Separate Classifier" and "Calculate Mapping".

## 3  RESULTS

To evaluate how these methods performed at predicting source dependent labels, we employed two public datasets. **MNIST** (LeCun et al., 2010); a collection of images of handwritten digits, and **PTB-XL** (Wagner et al., 2020; A et al., 2000); a collection of ECG measurements and a label corresponding to normal or abnormal heart behaviour (Section A.2 and A.3). The predictive model for MNIST and PTB-XL was a Multi-Layer Perceptron and 1D ResNet respectively (Section A.4).

Table 1: Accuracy (standard deviation) % on MNIST with an MLP model.

| Model | Number of Sources | | | | |
|---|---|---|---|---|---|
| | 5 | 10 | 50 | 100 | 500 |
| Input Append | 94.2 (0.4) | 93.1 (0.4) | 86.4 (0.6) | 68.7 (6.8) | 11.3 (0.1) |
| Classifier Append | 94.1 (1.0) | 79.0 (5.3) | 26.4 (0.9) | 17.7 (3.9) | 11.0 (1.0) |
| Predict Mapping | 94.4 (1.2) | 75.9 (5.6) | 30.9 (3.4) | 21.2 (1.1) | 13.2 (0.5) |
| Separate Classifier | **96.5 (0.2)** | **96.3 (0.3)** | **95.1 (0.8)** | **94.9 (0.3)** | **90.5 (0.9)** |
| Calculate Mapping | 93.5 (3.2) | 93.2 (3.3) | 70.3 (5.4) | 75.4 (10.0) | 63.4 (5.1) |

Table 2: Accuracy (standard deviation) % on PTB-XL with a ResNet model.

| Model | Number of Sources | | | |
|---|---|---|---|---|
| | 5 | 10 | 50 | 100 |
| Classifier Append | 60.0 (11.9) | 60.4 (5.7) | 51.3 (1.5) | 50.5 (1.1) |
| Predict Mapping | 77.2 (7.0) | 64.8 (5.6) | 56.8 (5.0) | 52.2 (1.8) |
| Separate Classifier | 84.1 (1.0) | 77.9 (11.9) | 79.8 (13.3) | 78.2 (12.7) |
| Calculate Mapping | **84.3 (2.0)** | **84.1 (0.9)** | **83.4 (1.8)** | **83.0 (1.5)** |

The "Separate Classifier" performed the best on MNIST (Table 1), with the least drop in performance as the number of sources increased. The performance of "Calculate Mapping" did not scale well with the number of sources in this experiment. On PTB-XL (Table 2), "Calculate Mapping" performed best, and scaled well when the number of sources was increased.

## 4  CONCLUDING REMARKS

Models capable of performing consistently and reliably across different data sources are important as they can enable more fair and robust predictions on real-world data. In this preliminary, proof-of-concept study, we suggest multiple methods for adapting predictive models to work on datasets in which label assignments are dependent on the source generating a data point. We found that creating separate classifier heads per source, and calculating a mapping from predictive model output to source predictions both performed well on different datasets. Future work will focus on developing methods further and researching how mappings can be learned directly from the source embedding.

URM STATEMENT

The authors acknowledge that at least one key author of this work meets the URM criteria of ICLR 2023 Tiny Papers Track.

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

# A APPENDIX

## A.1 PARALLELS WITH MULTI-TASK LEARNING

Comparisons can be drawn between our work and previous literature on multi-task learning. Within this field, machine learning methods are designed for applications in which data from the same domain may contain multiple label types (Zhang & Yang, 2022). Our problem setting can be viewed as a feature homogeneous multi-task environment, in which the feature spaces across tasks are consistent and data sources are viewed as separate tasks. Additionally, part of our method would be considered a feature transformation approach, in which the goal is to learn a feature representation of the data that is helpful across tasks. To see this, the deep-learning predictive model within all of our tested approaches discussed in Section 2 can be viewed as a feature transformation function, that maps input features to a latent space that is then used for task dependent predictions. There are multiple previous works that discuss the use of parameter sharing in neural networks across tasks (Liu et al., 2015; Mrkšić et al., 2015; Zhang et al., 2020) for the learning of consistent feature spaces, similar to our approach referred to as "Separate Classifier". However, the method referred to as "Calculate Mapping" employs further techniques from linear algebra not present within the literature from Multi-Task Learning.

Within our research, we are focused on the scenario in which the label-space across tasks is consistent, and in which label assignments across tasks are combinations of each other. A similar setting was discussed in Quadrianto et al. (2010) (where label-spaces are similar, but might not be compatible), in which the authors maximised mutual information between label sets to build a classifier capable of making predictions across tasks. However, this work trained separate models for each label set, optimised jointly, and is not directly applicable to deep learning models. Future work however, could explore the maximisation of mutual information between label sets in combination with deep learning models. This would involve optimising models for predictions on individual label sets jointly with the mutual information between predictions.

## A.2 THE DATASETS

**MNIST** (LeCun et al., 2010): A collection of 70,000 images of handwritten digits of resolution $28 \times 28$.

**PTB-XL** (Wagner et al., 2020; A et al., 2000): A collection of 21837 ECG measurements using 12 leads sampled at 100Hz on 18885 patients, and a label corresponding to normal or abnormal heart behaviour. Each data point represented 10 seconds of measurements.

## A.3 EXPERIMENTAL SETTING

To simulate multiple data sources, each containing unique label assignments, we first randomly split the dataset into the given number of sources, ensuring no data point is in two different sources.

Second, for each source we build a label mapping by randomly assigning a new label to each of the previous labels and transform all of the labels within each source. Next, we split the data in each source into a training and testing set which is used for model training and evaluation. This simulates the setting in which a machine learning model is being deployed on data in which inputs are consistent, but label assignments across sources present within the data differ.

Consequently, a proposed solution model will have learnt representations of the input useful for inference in all sources, whilst also being able to make source dependent predictions.

This work is applicable to any scenario in which the same input data could have different label assignments across data sources. Some brief examples include:

- Recommendation systems, in which an item with equivalent attributes might be more or less applicable to different people.
- Healthcare activity monitoring, where similar daily activity patterns might be normal or abnormal depending on a given patient.
- Personalised predictive modelling, in which personal differences would lead to different labels assigned to the same event.

## A.4  MODEL ARCHITECTURES AND HYPER-PARAMETERS

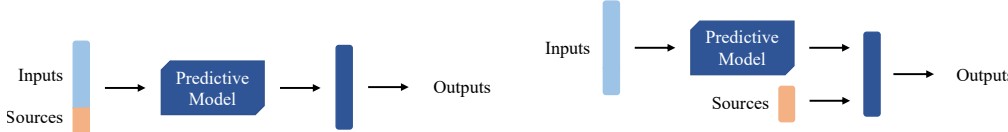

(a) **Input Append:** We include the source information in the input.

(b) **Classifier Append:** We append the source information to the output of a predictive model and apply another fully connected layer before the output.

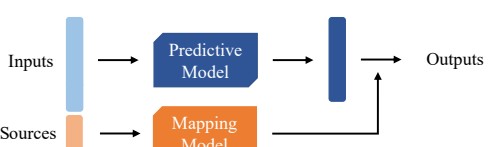

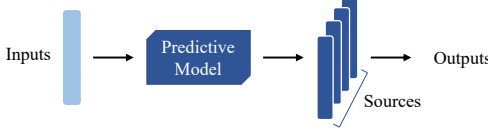

(c) **Predict Mapping:** We use the source information to predict a binary matrix that maps the outputted layer from the predictive model to a source dependent output.

(d) **Separate Classifier:** A separate classification layer is built per source. **Calculate Mapping:** A mapping matrix is calculated per source, to allow for the mapping of the output from a predictive model to the source assigned labels.

Figure 2

**MNIST**: The predictive model shown in Figure 2 is a Multi-Layer Perceptron (MLP) with hidden layers of sizes [100, 100, 10]. The entire model was trained for 100 epochs, with an early stopping criterion on the validation loss with a patience of 5 and a tolerance of 0.0001. It was optimised using Adam (Kingma & Ba, 2014) with a learning rate of 0.001, $(\beta_1, \beta_2) = (0.9, 0.999)$, and a batch size of 512. When calculating the mapping per batch, we used a weighted sum in which the newly calculated mapping was weighted 5:1 compared to the previous mapping. These parameters were chosen using a grid search.

**PTB-XL**: The predictive model shown in Figure 2 is a 1 dimensional ResNet (He et al., 2016) with 4 blocks, each increasing the number of features by 12 and decreasing the number of output time steps by a factor of 4. Since the input had 12 features and 1000 time steps, the final output from the ResNet blocks had 60 features and 3 time steps, which when flattened, was passed to a hidden layer that outputted 2 features, for the binary prediction task. The entire model was trained for 100 epochs, with an early stopping criterion on the validation loss with a patience of 5 and a tolerance of 0.0001. It was optimised using Adam (Kingma & Ba, 2014) with a learning rate of 0.001, an L2 parameter of 0.0001, $(\beta_1, \beta_2) = (0.99, 0.999)$, and a batch size of 1024. When calculating the

mapping per batch, we used a weighted sum in which the newly calculated mapping was weighted 1:1 compared to the previous mapping. These parameters were chosen using a grid search.

All experiments were run on random permutations of the dataset, and with new random model parameter initialisations 5 times to ensure reliability.

### A.4.1 CALCULATING THE MAPPING

The proposed solution "Calculate Mapping", involves calculating a change of basis matrix to map from outputs of the predictive model (seen in Figure 1b) and the source dependent labels. To start, the label-space mapping for all sources was initialised as the identity matrix. Then, the outputs of the predictive model are treated as predictions in their own label-space (the union of the possible labels in all source label-spaces) by applying the softmax function and calculating the label associated with the maximum value. For all source and predicted label combinations in a given set of outputs, the distribution of true labels is calculated. A weighted mean of these distributions, along with the current set of mapping matrices is calculated and chosen as the new mapping matrices. Before these mapping matrices are used to change the basis of the label-space in the predictive model, the maximum value for each source and predicted label combination is chosen, and this corresponding label in the source label-space is chosen as the mapped label. This ensures that each label in the predictive model label-space is only mapped to a single label in the source label-space.

The updating of the mapping matrix as described can be done at any interval, however within this work we chose to calculate the new mappings every batch, after the update gradients of the predictive model had been used in backpropagation.

