# OpenReview forum: "Predicting Targets with Data from Non-Conforming Sources "
_ICLR.cc/2023/TinyPapers — Submitted to Tiny Papers @ ICLR 2023_

### Official Review · Reviewer_VPuR · 2023-03-27

**Confidence:** 4

**Summary Of Contributions:**

The paper presents methods for building classification models that can make predictions on data with label assignments that depend on the data source. The authors explore this less explored area of importance and evaluate multiple methods such as Input Append, Classifier Append, Predict Mapping, Separate Classifier, and Calculate Mapping for learning multiple label assignments dependent on a point’s underlying data source.

**Rating:**

Great Start (GS): a submission which meets some of the reviewing criteria but has room for improvement

**Strengths And Weaknesses:**

Strengths:
- The paper addresses an important and less explored problem of making predictions on data generated by multiple sources.
- The paper proposes multiple methods for learning multiple label assignment that depend on the data source.
- The paper provides an evaluation of these methods on two public datasets, which adds credibility to the preposed methods.

Weakness:
- What is the source of MNIST and PTB-XL? Did you divide the training set into multiple sources, and set the label space differently for each source in the experiment? There should be an explanation of the experimental setup.
- Is it correct that a person manually calculates the mapping matrix in "calculate mapping"? It would be helpful to have an explanation.

**Suggested Changes:**

Suggested changes:

- In classification, the mapping matrix for change of basis should be one-hot per column. It would be good to try representing the output of the mapping model in the “predict mapping” method as one-hot.
- The cell labeled “Model” in the tables and the empty cell below it would look nicer if merged using a LaTeX `Multirow' package.
- "calculate mapping" and "calculated mapping" are being used interchangeably. Please unify them.
- I think the context of the statement that "$C_{S_i}$ can be written as an identity matrix" is not clear. It would be helpful to clarify the statement by indicating the dimensions of $M_{S_i}^{S_j}$ and $C_{S_i}$ and providing a clear explanation.

---

> ### Author Response · Authors · 2023-04-28
> **Reviewer Response**
>
> We thank the reviewer for taking the time to consider this work and for their constructive comments. We have updated the manuscript with further detail related to the questions here.
>
> - We have included a more detailed explanation of the experimental setting, in Appendix A.3, which the reviewer might find of interest.
> - The mapping is calculated based on outputs made by the predictive model, and the true labels given in each of the source assigned label spaces. The manuscript now contains a more detailed explanation, given in Appendix A.4.1. Please do let us know if you have any questions on this and I will update the work to make it more clear. Code will also be made available after review for further inspection.
> - We have added a note to clarify that the “Predict Mapping” mapping matrix is binary, and ensuring that it is one-hot would be an interesting addition for future work.
> - Thank you for the table layout suggestion, we have implemented the change.
> - "Calculate Mapping" and "Calculated Mapping” are now unified.
> - We have added the dimensions of the matrix to the description, and made it clear that these matrices are one-hot encodings of the labels. Additionally, we have added that the identity matrix can therefore be constructed by rearranging the columns of the matrix containing the basis vectors.
>
> We have also referenced the additional appendices in the main body of the manuscript.
>
> We are glad that the reviewer agrees with us that this work is an interesting avenue of research and begins to tackle an important problem in machine learning. Please let us know if you have any further questions or recommendations.

---

### Author Response · Authors · 2023-05-31
**Opt-in for Archive**

I wish to opt-in for archival

---

### Meta-Review · Area_Chair_SgxS · 2023-04-06

**Recommendation:** Invite to archive
**Confidence:** 4

**Metareview:**

Strengths:
- The paper studies a relevant and timely research problem, that of learning with data coming from multiple sources.
- The presentation of the work is good, and proposed concepts are nicely illustrated.

Weaknesses:
- Misses connection with important multi-task and multi-domain learning literature, where related techniques of encoding sources and cross-task mappings have been studied.
- The experiments seem to use random subsets of data, so it is not clear what is learned by the techniques.

**Summary:**

The paper considers the problem of learning with multiple different data sources, and proposes methods to predict while taking the source into account. The problem is very interesting and important, but the proposed approaches are not connected with the literature and the experiment methodology could be improved.

**Comments And Feedback To The Authors:**

- The proposed approaches seem very similar to techniques frequently used in multi-task and multi-domain learning (for example, [1] and [2]). It would be good to try to find them in the literature and cite/compare with the currently proposed methods.
- Using random permutations of datasets does not seem ideal to compare the usefulness of proposed approaches as it is not clear what information would be contained in each "source" that is being learning. Consider using data sets where there are multiple sources or tasks.


[1] Thrun, S., & Pratt, L. (Eds.). (2012). Learning to learn.

[2] Yang, Y. and Hospedales, T.M., 2014. A unified perspective on multi-domain and multi-task learning.

**Reason For Not Giving A Higher Recommendation:**

- A main concern is correctness of experimental methodology since random data subsets are being used.
- Another important issue is lack of connection to multi-task/domain learning literature, where similar approaches have been extensively studied.

**Reason For Not Giving A Lower Recommendation:**

The paper has good presentation and details of the approaches.

---

> ### Author Response · Authors · 2023-04-28
> **Reviewer Response**
>
> Thank you for taking the review our manuscript and provide constructive feedback. To address your comments, we have updated the manuscript with further information:
> - On the connection our work has with Multi-Task Learning, we have added a section in Appendix A.1 that discusses the parallels our work has with previous literature on learning from multiple tasks.
> - Since this is a proof-of-concept study, we wanted to use publicly available synthetically produced data to ensure results were reproducible and we had fine-grained control of the experimental setup. Further experiments, on real world data will be important for future work, and we have added a discussion to Appendix A.3 which suggests some simple examples of scenarios in which this experimental setting could arise.
> We have also referenced the additional appendices in the main body of the manuscript.
>
> Once again, we thank the reviewer for their decision to invite this paper for archive and are pleased that they share our opinion that it is an interesting and important avenue of research. Please let us know if you have any further questions or recommendations.

---

### Decision · Program_Chairs · 2023-04-08

Invite to archive